# Protein Analysis of *A. halleri* and *N. caerulescens* Hyperaccumulators When Exposed to Nano and Ionic Forms of Cd and Zn

**DOI:** 10.3390/nano12234236

**Published:** 2022-11-28

**Authors:** Valentina Gallo, Valentina M. Serianni, Davide Imperiale, Andrea Zappettini, Marco Villani, Marta Marmiroli, Nelson Marmiroli

**Affiliations:** 1Department of Chemistry, Life Sciences and Environmental Sustainability, University of Parma, 43126 Parma, Italy; 2Experimental Station for the Food Preservation Industry—Research Foundation, 43121 Parma, Italy; 3Institute of Materials for Electronics and Magnetism (IMEM), National Research Council (CNR), 06128 Parma, Italy; 4The Italian National Interuniversity Consortium for Environmental Sciences (CINSA), 43124 Parma, Italy

**Keywords:** proteomics, 2D SDS-PAGE, nanoscale and ionic metals, quantum dots, hyperaccumulator

## Abstract

Hyperaccumulator plant species growing on metal-rich soils can accumulate high quantity of metals and metalloids in aerial tissues, and several proteomic studies on the molecular mechanisms at the basis of metals resistance and hyperaccumulation have been published. Hyperaccumulator are also at the basis of the phytoremediation strategy to remove metals more efficiently from polluted soils or water. *Arabidopsis halleri* and *Noccea caerulescens* are both hyperaccumulators of metals and nano-metals. In this study, the change in some proteins in *A. halleri* and *N. caerulescens* was assessed after the growth in soil with cadmium and zinc, provided as sulphate salts (CdSO_4_ and ZnSO_4_) or sulfide quantum dots (CdS QDs and ZnS QDs). The protein extracts obtained from plants after 30 days of growth were analyzed by 2D-gel electrophoresis (2D SDS-PAGE) and identified by MALDI-TOF/TOF mass spectrometry. A bioinformatics analysis was carried out on quantitative protein differences between control and treated plants. In total, 43 proteins resulted in being significatively modulated in *A. halleri*, while 61 resulted in being modulated in *N. caerulescens*. Although these two plants are hyperaccumulator of both metals and nano-metals, at protein levels the mechanisms involved do not proceed in the same way, but at the end bring a similar physiological result.

## 1. Introduction

One group of nanoparticles extensively exploited for their biomedical and industrial applications is Quantum dots (QDs). A characteristic the Quantum dots is a metal core composed of cadmium (Cd), tellurium (Te), arsenic (As), lead (Pb), zinc (Zn), copper (Cu), or selenium (Se), alone or in combination [1]. The use of engineered QDs is advantageous because they have a small size, regular crystal shape, a large surface area to mass ratio, and possess strong affinity to metals, thus reducing their availability for plants in metal-rich soil. Plants in natural environments are constantly exposed to a combination of biotic and abiotic stress [2]. Drought, salinity, water logging, heat, cold, metals, and UV radiation are hurdles which plants face at some degree of their life cycle [3]. In agricultural soils, an excessive level of metals can reduce soil quality, crop yield, and compromise the integrity of agricultural products, as well as cause possible hazard to humans, animals, and ecosystem health. On metal-rich soil, certain plant species and varieties have evolved and adopted toxic metals as common minerals [4]. Metal hyperaccumulators have developed an exclusive strategy to achieve tolerance by diverting the largest amount of metal into the aerial parts, stems, and leaves with the formation of complex coordination bound either with 0 or S, which makes these complex less toxic [5].

Hyperaccumulator plants could become a resource for phytoremediation of metals [6,7] to extract metals from soil and concentrate in their upper parts. Several species of the *Brassicaceae* family are metal hyperaccumulators [8], and studies have suggested an evolutionary role of hyperaccumulation as a defense against the herbivores [9] and pests [10]. *Noccaea caerulescens* (former *Thalaspi caerulescen*) and *Arabidopsis halleri* have been model plants in metal hyperaccumulation [7,11], largely because of their genetic analogy with the model species *Arabidopsis thaliana* (L.) Heynh and because their genomes have 94% and 88.5% correspondence in coding regions, respectively [12]. *N. caerulescens* is a Zn and Cd hyperaccumulator and accumulates these metals at concentrations exceeding 1% of shoot dry biomass; Cd seems to be less consistently hyperaccumulated than Zn [13]. Similarly, *A. halleri* is a known Cd hyperaccumulator but also a Zn hyperaccumulator [14]. The capacity that some *Brasicacee* family members have for storing large amounts of divalent cations in their vacuoles might explain why these families comprise about 50% of all know metal hyperaccumulation plant taxa (about 400 specie) [15]. While hyperaccumulators are “competitive” in the presence of high metals, they do not suffer particular disadvantages when the environment conditions are non-selective, showing a large degree of phenotypic plasticity [16]. A way to understand the plasticity of the hyperaccumulator phenotype is considering the interaction between gene and environment at a protein level. A proteomic approach can show which proteins were involved in the response to metal-based NMs like QDs. Recently, it was suggested that cases of environmental dispersion of NMs could be treated with plants as in phytoremediation of metals [17]. Here, we suggest that QDs phytoremediation can benefit from the knowledges emerging from proteomic study of two hyperaccumulator models, *A. halleri* and *N. caerulescens*, in terms of functions and genes useful for implementing the technology.

## 2. Materials and Methods

The ZnS QDs and CdS QDs (uncoated) were synthesized by CNR-IMEM (Parma, Italy) following the methods of Villani et al., (2012) [18]. The CdS QDs have a bulk density of 4.82 g cm^−3^ and an average diameter of 5 nm. Cd represents 78% of the dry weight of the QD; the *Z*-potential for these nanoparticles was +61.6 mV and they had a hydrodynamic range of 1190 nm. The ZnS QDs had an average diameter under 5 nm, with Zn representing 63% of the dry weight of each QD. The ZnS QDs have a *Z*-potential of +34.9 mV and a hydrodynamic range of 248.7 nm. Details of the synthesis and characterization are provided in Imperiale et al. (2022) [19].

### 2.1. Growth Condition and Treatments

Seeds of *A. halleri* Langhelsheim were kindly provided by Prof. S. Clemens (Department of Plant Physiology, University of Bayreuth, Germany). Seeds of *N. cearulescens* Monte Prinzera accession were taken from our collection from the Monte Prinzera site (Fornovo, Parma, Italy), a serpentine hill near Parma, Italy. One seed of *A. halleri* and *N. caerulescens* was placed in each pot and was germinated in a growth chamber (MIR-554-PE, Panasonic, Osaka, Japan) for 72 h at 25 °C, under a relative humidity of 50% and dark conditions. After germination, pots with seedlings were transferred to a greenhouse with average temperatures of 25/16 °C (day/night), and light-supplemented by metal halide lamps to maintain a minimum light intensity of 300 μmol m^−2^ s^−1^, with a 14 h photoperiod. According to their growth cycles, plants were grown for different periods prior to treatment: 60 days for *A. halleri* and 45 days for *N. caerulescens*. The watering frequency was 30 mL of deionized water, twice a week. After growth, plants were treated: *A. halleri* with 10 mg of Cd administered as CdSO_4_ and CdS QDs and *N. cearulescens* with 10 mg of Zn administered as ZnSO_4_ and ZnS QDs; three biological replicates for condition (Appendix A). The plants were maintained for an additional 30 days under the same conditions of temperature, humidity, light, and irrigation. One gram of the aerial part of plant material was collected for each condition, gently washed with distilled H_2_O, frozen in liquid nitrogen, and then stored at −80 °C for the following proteomic analysis. The controls (untreated) were prepared in the same way. The experiment was conducted in triplicate.

### 2.2. Proteins Extraction and Separation with Two-Dimensional Gel Electrophoresis (2D SDS-PAGE)

Total proteins were extracted from leaves of *A. halleri* and *N. caerulescens*, grown under the experimental conditions previously described (Appendix A). The proteins extraction was performed with a phenol-based buffer as described in Gallo et al. (2021) [20]. Protein’s concentration was evaluated according to a modified Bradford assay based on the acidification of the sample [21] buffer with 20 mM HCl. Bovine serum albumin (BSA) was used as standard.

The selected pH gradient must compose all pI of proteins in the mixture and allow for the best separation for maximal resolution of gels. In this case, the strip with 5–8 pH range was selected for *A. halleri* and the strip with 3–10 pH range was selected for *N. caerulescens* because it allowed the detection of the larger number of spots and was used throughtout for all samples. This decision to use the range of pH 5–8 for *A. halleri* was taken after the paper by Farinati et al. (2009) [22], because they used a pH range of 3–10, but it was clear that most of the proteins were in the range 5–8. For protein separation in the first dimension, 300 µg of total proteins from each sample were loaded onto 11 cm ReadyStrip pH 5–8 and 3–10 IPG strips (BioRad, Hercules, CA, USA) which were rehydrated overnight with 250 µL IEF buffer containing the sample. Proteins were focused by using the PROTEAN^®^ i12™ IEF System (BioRad, Hercules, CA, USA) following this program: 250 V (60 min), 1000 V (60 min), 8000 V (2 h) to reach 8000 V for a total of 35,000 V/h. After IEF, the strips were incubated 15 min in 3 mL of reducing buffer containing 2% *w*/*v* DTT, 6 M Urea, 0.375 M Tris-HCl (pH 8.8), 20% *w*/*v* glycerol, 2% *w*/*v* SDS, and for 15 min in 3 mL of alkylating buffer containing 2.5% *w*/*v* iodoacetamide, 6 M Urea, 0.375 M Tris-HCl (pH 8.8), 2% *w*/*v* glycerol. The second dimension (SDS-PAGE) was performed using a CriterionTM Dodeca™ cell (BioRad, USA) and 12% Criterion™ XT Bis-Tris gels (BioRad, USA) in 1 M MOPS (3-(N-morpholino)-propanesulfonic acid) buffer containing 1 M Tris, 20 mM EDTA, and 2% *w*/*v* SDS.

2D gels were stained with QC Colloidal Coomassie G-250 (BioRad, Hercules, CA, USA) and gel images were obtained with a ChemiDocTM Imaging System (BioRad, USA). Image analysis was performed using the PDQuest software (BioRad, Hercules, CA, USA). Spots detection and matching between gels were performed automatically, followed by manual verification. The spots densities were normalized by local regression method and subsequently against the whole gel densities. The percentage density of each spot was averaged over three replicates’ gels and Student’s t-test analysis (*p* < 0.05) was performed to find out statistically significant differences in proteins abundance. Statistically relevant spots were successively excised from the gels using an EXQuest Spot Cutter (BioRad, Hercules, CA, USA), distained by soaking the pieces of acrylamide for 30 min in a 1:1 solution of 100 mM ammonium bicarbonate and acetonitrile, and the proteins were hydrolyzed with trypsin following Shevchenko et al.’s (2006) protocol [23]. 

### 2.3. Protein Identification and Data Mining

Tryptic peptides were desalted and concentrated to a final volume of 4 µL with Zip-Tip C18 (Millipore Corporation, Billerica, MA, USA), according to the manufacturer’s protocol, then dispersed into an α-cyano-4-hydroxycinnamic acid (4-HCCA) matrix, prepared by dissolving 4-HCCA in 50% acetonitrile/0.05% trifluoroacetic acid, and spotted on a MALDI plate. The samples were subjected to analysis through a model 4800 MALDI-TOF/TOFTM MS analyzer (Applied Biosystems, Foster City, CA, USA). Peptide mass fingerprint (PMF) were acquired in reflectron mode (500–4000 *m*/*z* range) and analyzed with the help of mMass v5.5 open-source software (http://www.mmass.org/; accessed on 10 October 2020). For each feature, a peak list was created and then manually checked for the presence of signal from the matrix complex, trypsin, and human keratin peptides. PMF analysis was carried out with the software Mascot (http://www.matrixscience.com, last accessed November 2020) and proteins were identified by searching for confidence of match in the Swiss-Prot database (Swiss-Prot Viridiplantae taxonomic subdatabase of ‘nr’nonredundant) of the UniProtKB-Swissprot.

### 2.4. Statistical Analysis, Data Mining, and Bioinformatic Analysis

Venn diagrams were constructed using the software Venny 2.1 program (Oliveros, J.C., 2007–2015, Venny http://bioinfogp.cnb.csic.es/tools/venny/index.html) last accessed the 1st of November 2022. The Gene Ontology analysis was performed through the Panther database (pantherdb.org/). Classification of proteins was performed using MapMan software (http://mapman.gabipd.org/mapman-download last accessed the 1st of October 2022) based on *A. thaliana* genome database and the results were manually verified using the annotations of UniProt database. MapMan was used to place proteins within a likely pathway (BIN). Heat maps of selected proteins were generated by R v3.3.1 (www.r-project.org, last accessed the 1st of November 2022).

## 3. Results

### 3.1. Plant Growth

In a previous paper, it was shown that samples of *Noccea caerulescens* from Monte Prinzera varied according to the subsites and may give plants with different phenotypes according to the subsite [24]. The plants we used were taken from the same subsite, number 3, as were those in the paper from Imperiale et al., (2022) [7], where a thorough description is provided. These plants, as those of *Arabidopsis halleri*, not only grow well in the presence of Cd and Ni at high concentrations, which may affect negatively other plants’ non-hyperaccumulators, but also, they seem to need high concentrations of metals to grow more vigorously, as shown in Appendix A.

### 3.2. Differential Proteomic Analysis: Comparison between Plants and Treatments

After acquisition by Bio-Rad’s PDQuest 2-D analysis software, the gels were analyzed. A number of 206 and 180 spots were sufficiently clear to be appreciated in *A. halleri* and *N. caerulescens*, respectively. The next step was comparing the spot intensities through comparative pairwise protein analysis between control, treated sample with QDs (CdS QDs and ZnS QDs), and treated sample with salts (CdSO_4_ and ZnSO_4_). Spots whose intensity resulted in being statistically significant using the *t-Student* test were considered (values below the threshold 0.05 were considered significant), and a number of 43 for *A. halleri* and 61 for *N. caerulescens* significative spots were identified whose intensities increased or decreased during the treatments (Figure 1). For *A. halleri*, the number of spots with significantly increased or decreased intensity are reported in Figure 1A as a Venn diagram. Among these 43 spots, 30 of these had an identification from the comparison between control and treatment with CdS QDs. The spots up-regulated with the treatment with CdS QDs were 12, while those down-regulated with the same treatment were 18. Comparison between control and treatment with CdSO_4_ identified 17 spots as statistically significant. The spots up-regulated were 3, while those that were down-regulated were 14 (Figure 1A). Spots shared among pairwise comparisons corresponded to proteins up- or down-regulated in the two different conditions and were highlighted by the intersections in the Venn Diagrams (Figure 1A). The spots shared between CdSO_4_ down and CdS QDs down were (DNA repair protein recA homolog 2, Pentatricopeptide repeat-containing protein At1g28690, Transcription factor KAN4, Ras-related protein RABE1d) (Figure 1A) all down-regulated.

Similarity, among the 61 spots of *N. caerulescens*, 42 of these were identified from the comparison of control and treatment with ZnS QDs, and 36 were identified from the comparison between control and treatment with ZnSO4 (Figure 1B). Among the 42 spots 23 were up-regulated, while those down-regulated were 19. From the treatment with Zn, among the 36 spots 17 were up-regulated, while those down-regulated were 19. Spots shared among pairwise comparisons corresponded to proteins up- or down-regulated in the two treatments and were highlighted by the intersections of the Venn Diagrams (Figure 1B). The spot shared between ZnS QDs down and ZnSO_4_ down is 1 (Alpha-1,6-mannosyl-glycoprotein 2-beta-N-acetylglucosaminyltransferase), whereas the spots shared between ZnS QDs up and ZnSO_4_ up were 7 (30S ribosomal protein S19 chloroplastic, 3-ketoacyl-CoA synthase 12, Cytochrome b-c1 complex subunit 7-2, Jasmonate O-methyltransferase, Protein trichome birefringence-like 6, Ras-related protein RABA1c, tRNA-splicing endonuclease subunit Sen2-1). There are also spots with opposite behavior (up- or down-regulated) compared to the treatments: the spots shared between ZnSO_4_ up and ZnS QDs down were 7 (2S seed storage protein 4, Late embryogenesis abundant protein 46, NADH dehydrogenase [ubiquinone] 1 beta subcomplex subunit 7, Nascent polypeptide-associated complex subunit beta, NDR1/HIN1-like protein 26, Serine carboxypeptidase-like 52, Thioredoxin F-type chloroplastic), whereas the spots shared between ZnSO_4_ down and ZnS QDs up were 2 (Defensin-like protein 39, and E3 ubiquitin-protein ligase PRT1) (Figure 1B). 

Proteins identified through comparative pairwise among three conditions in two types of plants are indicated in the tables (Appendix A). From the comparison of the two sets of proteins isolated and identified from two hyperaccumulators, it was evident that there was only a limited overlapping. In fact, there were only 3 proteins commonly modulated in the two hyperaccumulators (Figure 1C): ATP synthase delta chain (Atpd), Ribulose bisphosphate carboxylase large chain (rbcl), and F-box protein (At3g18320).

A heat map was used for a graphical representation of the protein abundance in treatments as compared to control, through the “R” software (*R Project for Statistical Computing*). The binary logarithm of the spots color intensities ratio between a treated sample and control was used to generate the heat map, consequently in the graphical representation, the values are represented by different shades of colors green and red. The columns represent two different treatments, while the rows represent the identified proteins considered for the comparison (Figure 2 and Figure 3). In *A. halleri*, Atpd resulted in being up-regulated in the treatment with QDs and down-regulated in the treatment with Cd salt, while in *N. caerulescens* resulted in being down-regulated in both treatments. In *A. halleri* Rbcl resulted in being down-regulated in both treatments, while *N. caerulescens* resulted in being up-regulated in the treatment with QDs and down-regulated in the treatment with Zn salt. In *A. halleri*, At3g18320 resulted in being up-regulated in the treatment with QDs and down-regulated in the treatment with salt, while *N. caerulescens* resulted in being down-regulated in both the treatments (Figure 2 and Figure 3).

### 3.3. Gene Ontology Analysis

PANTHER database (Protein ANalysis THrough Evolutionary Relationships) was used for the Gene Ontology analysis. Using information about genome sequencing of other plant organisms, three independent ontologies were used to define complete information for each inserted gene and to describe a genic product: molecular function, biological process, and cellular component. For *A. halleri*, the list of genes inserted in the software were 30 genes obtained from the comparative analysis between control and treated with CdS QDs and 17 genes obtained from the comparative analysis between control and treated with CdSO_4_. For *N. caerulescens*, the list of genes inserted in the software were 42 genes obtained from the comparison between control and treatment with ZnS QDs and 36 from the comparison between control and treatment with ZnSO_4_. Figure 4, Figure 5 and Figure 6 represented the common and uncommon molecular functions, biological processes, and cellular components in the two hyperaccumulators.

For molecular functions, the most represented class in the two comparisons was the catalytic activity (Figure 4). The common molecular functions in the comparison between Ctrl vs. QDs were catalytic activity, binding, transporter activity, and ATP-dependent activity. The exclusive classes of *N. caerulescens* were molecular adaptor activity and structural molecule activity (Figure 4A). The common classes in the comparison between Ctrl vs. salt were catalytic activity, transporter activity, transcription regulator activity, and binding, while the exclusive classes of *N. caerulescens* were molecular function regulator and structural molecule activity (Figure 4B). 

The percentage of genes that coded for proteins that have a specific biological function, involving multiple molecular functions, are reported in two different diagrams (Figure 5). In these diagrams, the most represented class, in the two comparisons, was cellular process. The commonly represented classes in a comparison between Ctrl vs. QDs were cellular process, metabolic process, localization, biological regulation, and developmental processes. The exclusive classes of *N. caerulescens* were growth and response to stimulus (Figure 5A). The common classes in a comparison between Ctrl vs. salt were cellular process, metabolic process, biological regulation, and localization. The exclusive class of *A. halleri* was a biological process involved in interspecies interaction between organisms and growth. 

The percentage of genes that coded for proteins that have a specific localization at the subcellular level or in macromolecular complexes is reported in Figure 6 (Figure 6). The common classes in the comparison between Ctrl vs. QDs were intracellular anatomical structure, cytoplasm, membrane, organelle, organelle subcompartment, endomembrane system, protein-containing complex, intracellular anatomical structure, cytosol, and cell periphery. The exclusive classes of *A. halleri* were supermolecular complex and endoplasmic reticulum site (Figure 6A). The common classes in the comparison between Ctrl vs. salt were organelle, protein-containing complex, cell periphery, intracellular anatomical structure, membrane, and cytoplasm. The exclusive class of *N. caerulescens* was the endomembrane system, while the exclusive class of *A. halleri* was the supermolecular complex (Figure 6B). 

### 3.4. Pathway Analysis 

Metabolic pathway analysis was performed by submitting the Gene IDs of the proteins identified in two different plants to the MapMan server (http://www.gomapman.org, last accessed the 1st of October 2022) for *A. thaliana* to identify the pathways that were more represented. For all the proteins, the MapMan ontology BIN assignations are listed in Appendix A for *A. halleri* and in Appendix A for *N. caerulescens*. In *A. halleri*, the set of reprogrammed proteins was associated with the following major MapMan bins: RNA regulation of transcription, secondary metabolism, miscellaneous (misc.), protein post-transcriptional modification, protein degradation, and lipid metabolism (Figure 7). In *N. caerulescens*, the set of reprogrammed proteins was associated with protein degradation, RNA regulation of transcription, photosynthesis and photorespiration (PS), signaling, and cell (Figure 7). This last pathway, together with amino acid metabolism and development, is exclusive of *N. caerulescens,* while DNA synthesis, secondary metabolism, and transport are exclusive of *A. halleri* (Figure 7). 

## 4. Discussion

The study of the hyperaccumulators *A. halleri* and *N. caerulescens* suggests that when the plants were treated with 10 mg of Cd or Zn in the nano form (CdS QDs or ZnS QDs) or as salt (CdSO_4_ or ZnSO_4_), the response can be quite similar at morphological and physiological levels [7]. The characterization of protein response to treatments with Cd or Zn in nanoscale or ionic form, due to the low number of proteins resolved by the 2D-gel, cannot be considered a throughout proteomic analysis [20] because of the throughput of the method. However, this approach was used instead of performing transcriptomic analysis because of the higher relatedness that proteins molecules had to gene expression and to plant phenotype. On the other hand, for the proteomic analysis, the 2D SDS- PAGE technique was because of its simplicity, considering that the number and the intensities of the spots differentially abundant in control and treatments were well evident considering their relative’s high amount. Moreover, 2D SDS-PAGE is a ‘top-down’ proteomics method that analyzes intact proteins directly by preserving their post-translationally modified forms displayed in vivo without any previous proteolytic digestion [25]. As a prototypical ‘top-down’ proteomics technique, the 2D gel-based proteomics have relevant applications [26,27]. Although this technique has some advantages, such as the possibility of identifying post-translational modified (PTM) proteins and protein isoforms, it also has significant limitations [28]. The application here of this comparative analysis has allowed for identifying the spots with intensity values as statistically significant. Considering the type of staining used, the concentrations of proteins estimated for the spots selected were in the order of the ng. In this paper, we were searching for those major changes which can make sense to the phenotypic differences between the two species of plants which, at the end, converge to a similar evolutionary acquisition: metal tolerance and metal hyperaccumulation. The spots shared between treatments in the pairwise comparisons were displayed in a Venn Diagram as in Figure 1. The spots that were relevant from a biological point of view were then identified by MS and assigned to a specific molecular function, metabolic pathway, and cellular component. Considering the different origins of the plant species, from a laboratory seed collection for *A. halleri*, and from a natural site for *N. caerulescens*, and the different treatments administered, Cd and Zn, respectively, it was not surprising that there was a low overlap in the number of regulated proteins found. Besides, this result was already evident from the two papers [22,24].

The two different pH ranges used for the first-dimension gel in the two hyperaccumulators respected the separation patterns that we observed at all pH ranges tested. The range between the two was 5–8 for *A. halleri* and 3–10 for *N. caerulescens.* After a fine-tuning of the protocol, the same pH-range was used for *A. halleri* as for *A. thaliana* (pH 5–8) [20], as it respected the same protein profile. On the other hand, *N. caerulescens* seemed to have a more extensive protein profile towards the acidic pH part. For this reason, a more extensive pH range was used (pH 3–10).

### 4.1. Proteomic Analysis of Differentially Expressed Proteins in A. halleri

After the treatments with ionic or nanoforms *A. halleri*, Cd showed some specific proteomic features (Figure 2) as the downregulation of the ABC transporter protein (Abcb28), being involved in the transport of metals in the vacuole and in other cellular structures [29]. The stress induced by nanomaterial treatment has decreased the presence of proteins with antioxidant activity: aldo-keto reductase (Akr4c11), glutathione S-transferase (Gstu18), peroxidase (Per10), cinnamoyl-CoA reductase 2 (Ccr2). Particularly, GSTs were involved both in detoxification and in hyperaccumulation of Cd [30], and it was modulated both in the control and in the treated samples. However, the behavior of glutathione S-transferase (Gstu24) was opposite in the treatment with CdS QDs as compared to treatments with CdSO_4_, as highlighted in the heat map, whereas the other oxidoreductases were down-regulated after the two treatments (Figure 2). The downregulation of GSTFs by metals was also observed in another study, and requires further investigation because GSTs are usually induced under metal stress [30]. Similar evidence and a similar conclusion are reported in another paper [22].

Two transcription factors were down-regulated by the treatments: transcription factor CPC (Cpc) and KAN4 (Kan4) (Figure 2). The first is expressed in trichomes and in young developing leaves, and it represses trichome development by lateral inhibition [31]; the second regulates carpel integuments formation by modulating the content of flavonols and proanthocyanidin in seeds [32]. It is of particular importance to have found proteins related to the trichome because, in the literature, there is evidence that Cd in *A. halleri* is mainly stored in the leaf trichomes [33]. Moreover, Zhao et al., (2006) [34] found that *A. halleri* hyperaccumulate Cd and that the root-to-shoot translocation of Cd was in a certain way inhibited by Zn, hinting that Cd enters *A. halleri* cells partly through the Zn transport pathway. Schvartzman et al., (2018) proposed that transcription factors such as HMA4 were selected during evolution of *A. halleri* and contribute to metal tolerance by enabling metal storage in shoot tissues. In addition, there is evidence that metal tolerance protein 1 (Mtp1) is probably responsible for Zn storage in shoot vacuoles, attributed to Zn hyper-tolerance, but can also transport cadmium with a low efficiency [35]. The protein Mtp1, in this proteomic analysis, was up-regulated in two treatments of *A. halleri*, therefore, it seems important that we found this protein as proteins modulated in *A. halleri* by the treatments, transcription factors, and metal-chaperones that may serve for the translocation and the storage of the metals in the above parts of the plants.

On the contrary, the gene for heavy metal-associated isoprenylated plant protein 8 (Hipp08) was up-regulated in both treatments (Figure 2). Hipp08 is a metallochaperone protein for the safe transport of metallic ions inside the cell [36]. HIPPs may be involved in heavy metal homeostasis and detoxification, particularly concerning Cd tolerance, transcriptional responses to cold and drought, and plant–pathogen interactions [31]. It is evident that the treatment with CdS QDs determines the modulation, up or down, of proteins involved in oxidative phosphorylation as the ATP-dependent activity class listed in Figure 4A. In fact, Gallo et al., (2020) [37] demonstrated that CdS QDs treatments caused a reduction in oxidative phosphorylation and in ATP production in yeast. Moreover, some proteins involved in the energy production, ATP synthase subunit alpha (Atp1) and ATP-dependent Clp protease ATP-binding (Clpt1), were down-regulated (Figure 2). It is evident that the regulation imposed by ZnS and CdS QDs required more ATP, possibly because it is more complex [37,38]. Farinati et al., indicated a general upregulation of photosynthesis-related proteins in *A. halleri* exposed to metals Cd and Zn, and to metals plus microorganisms, suggesting that metal accumulation in shoots is an energy-demanding process [22]. In the biological process bar chart, two classes, represented in the comparison between Ctrl vs. salt, were absent in the comparison between Ctrl vs. QDs response to stimulus and biological process involved in interspecies interaction between organisms, while one class, development process, was present in Ctrl vs. QDs and was absent in Ctrl vs. salt (Figure 5A). The proteins identified in these classes were Transcription factor CPC and Cytochrome P450 82G1. Some of these class proteins, not necessarily those identified here, could also be involved in “biotransformation” of nanoparticles which takes place after nanoparticles internalization [39]. In the class biological regulation, in Ctrl vs. QDs, belong three proteins: GTP-binding protein Sar 1a, Thioredoxin-like 2-1, and Transcription factor KAN4. Sar 1a plays a key role in the control of vesicle transport and could be involved in the cellular transport of nanoparticles [40]. This protein was up-regulated in both treatments (Figure 2). Thioredoxin-like 2-1 is an oxidoreductase that may participate in various redox reactions and some assays of subcellular localization demonstrated that this protein is localized to the chloroplast, which is known to play a key role in the response to ionic and nanomaterials [38,41]. In the cellular components bar chart (Figure 6A), the membrane class emerges when comparing Ctrl vs. QDs. Delta-9 desaturase-like 2 protein (At1g06100), D-2-hydroxyglutarate dehydrogenase (D2hgdh), and ABC transporter B family member 28 (Abcb28) were among the more interesting. Abcb28 is involved in the transport of metals toward the vacuole or other cellular structures, and it is mostly responsible for the tolerance that hyperaccumulators have [29]. The same transport system was also involved in response to QDs, which favors the concept that vacuolar localization, stable or transient, is involved in the metals hyperaccumulation. At1g06100 the polyunsaturated fatty acid biosynthesis is involved [42] and was up-regulated in both treatments. Some membrane proteins indeed could be over-produced in the presence of nanoform of Cd and in the presence of ionic Cd.

### 4.2. Analysis of Differentially Expressed Proteins in N. caerulescens

Similarly to *A. helleri* and *N. caerulescens* after the treatments, the most affected proteins were involved in oxidative stress. In this case, the comparison concerned Ctrl, nano Zn, and ionic Zn. Probable glutathione S-transferase parC (ParC) was up-regulated in both treatments, peroxisomal membrane protein 11-1(Pex11-1) was down-regulated in the treatment with QDs and up-regulated in the treatment with salt, and E3 ubiquitin-protein ligase (Ari9) was down-regulated in both treatments (Figure 3). Four transcription factors were down-regulated by the treatments: transcription factor SRM1 (Srm1), transcription initiation factor TFIID subunit 15 (Taf15), heat stress transcription factor B-3(Hsfb3), and transcription factor IND (Ind) (Figure 3). A downregulation of these transcription factors suggests that there could have been an inhibition of transcription due to the treatments. In particular, the presence of the transcription regulator activity class in GO analysis is also highlighted (Figure 4B). Part of these proteins were already described in Visioli et al., (2012), though in a different context, because different extraction methods were used and the study was on metal ions only [24]. Concerning GO analysis, in the molecular function bar chart, the ATP-dependent activity class is shown in the comparison Ctrl vs. QDs (Figure 4A). Two proteins involved in energy production, ATP synthase delta chain chloroplastic (Atpd) and ATP-dependent Clp protease proteolytic subunit-related protein 1 chloroplastic (Clpr1), were down-regulated (Figure 3) as reported for *A. halleri.* These two proteins were localized together in the chloroplast, and previous studies in *A. thaliana* treated with QDs highlighted the role of chloroplast as a potential target of the ENMs [20,43]. Moreover, it is reported that mutation in a gene encoding for a protein of chloroplast function led to resistance to QDs [40]. Clpr1was required for chloroplast development and differentiation [44]. Atpd is an ATP synthase which produces ATP from ADP in the presence of a proton or sodium gradient and is essential for photosynthesis, facilitating electron transport in both photosystems I and II [45]. Wang et al., (2016) showed that CuO NPs impedes electron transport between the two photosystems which can cause an increase of ROS and generation of oxidative stress, damaging biological molecules and disrupting cellular metabolism [46]. Additionally, in the previous study [24], there was a report of several proteins involved in energy metabolism (both in chloroplast and mitochondrion) that were increased. Therefore, they can compensate for a reduction in ATP synthase.

### 4.3. Pathway Analysis

Mapman ontology allowed for the identification of several common pathways, particularly those involved in the response to nano or ionic form of Cd and Zn. Many proteins identified belong to RNA regulation of transcription bin. Interestingly, in *N. caerulescens* there were two proteins, Protein SHI related sequence 1(Srs1) and Zinc finger WIP3 (Wip3), which belong to the zinc family transcription factor (TF) family. Srs1 is a transcription activator and promotes auxin homeostasis by regulating gene expression as well as genes affecting stamen development, cell expansion, and timing of flowering [47]. Synergistically with other SHI-related proteins, it regulates gynoecium, stamen, and leaf development in a dose-dependent manner, controlling apical-basal patterning, promoting stilo and stigma formation, and influencing vascular development during gynoecium development [47]. This effect can extend its influence on flower development, which was also observed and described in Zucchini [39]. Wip3 belongs to WIP proteins, a plant specific subfamily of C_2_H_2_ zinc finger (ZF) proteins and works in order to coordinate zinc ions to maintain characteristic tertiary of the protein structures [48]. These two proteins, Srs1 and Wip3, were down-regulated after both treatments (Figure 3). Another important TF was Hsfb3, which belongs to heat stress transcription factors (Hsfs) family. Hsfs are the terminal components of a signal transduction chain mediating the activation of genes responsive to both heat stress and to chemical stressors [49]. In *A. halleri,* proteins involved in regulation of transcription, such as the AT-rich interactive domain-containing protein 5 (Arid5), Transcription factor CPC (Cpc), and Transcription factor Kan4 (Kan4), were identified.

Recent papers show that treatment with ENMs increases or decreases functionality of the organelles in *A. thaliana* [38,50]. In fact, in chloroplasts, the common protein ATP synthase delta chain (Atpd) in *A. halleri* was down-regulated in the treatment with CdSO_4_ and up-regulated in the treatment with QDs, whereas in *N. caerulescens* it was down-regulated after both treatments. This same protein had a different behavior in the two hyperaccumulators. Moreover, the same protein NAD(P)H-quinone oxidoreductase subunit K, localized in the chloroplast, was up-regulated after treatment with QDs and down-regulated after treatment with ZnSO_4_. Additionally, the presence of Ribulose bisphosphate carboxylase large chain (Rubisco subunit) corresponded to a higher rate of photorespiration. In fact, the photorespiratory pathway represents a way to recover C atoms ‘‘wasted’’ by this side activity of Rubisco. The cycle requires the coordinated activity of chloroplasts, peroxisomes, and mitochondria. In *N. caerulescens* there are three proteins that are part of the mitochondrial electron transport and ATP synthesis pathway: Ubiquinone biosynthesis protein COQ4 homolog (COQ4), Cytochrome b-c1 complex subunit 7-2 (QCR7-2), and NADH dehydrogenase [ubiquinone] 1 beta subcomplex subunit 7 (At2g02050). The first two proteins were up-regulated after both treatments, while At2g02050 was down-regulated in the treatment with ZnS QDs and up-regulated in the treatment with ZnSO_4_ (Figure 3). Conversely, in *A. halleri*, no proteins belonging to this pathway showed a particular regulation after treatments. These data, together with other present in literature, suggest that the hyperaccumulation of heavy metals in shoots of hyperaccumulator plants is energetically extremely costly, so the plants can afford this cost by reinforcing the photosynthetic mechanism modulating proteins involved in photosynthesis and those components that act as scavengers of the unavoidable by-products that occur with this extensive energy handling [22,24]. This phenomenon was also explained in terms of “trade-off” between mass growth and hyper-tolerance in [51]. 

### 4.4. Comparison between the Two Hyperaccumulators and A. thaliana

To make a comparison intra species with *A. thaliana* treated with CdS QDs, we took advantage of a similar study performed in *A. thaliana* [20]. A comparison of the three sets of proteins was identified: the two hyperaccumulators and *A. thaliana* showed only a limited overlapping. In fact, there was only 1 protein modulated in common among the three: the Ribulose bisphosphate carboxylase large chain (rbcl), while *A. thaliana* and *N. caerulescens* had tRNA-splicing endonuclease subunit Sen2-1 (Sen1) in common. Sen1 is located in the nucleus and constitutes one of the two catalytic subunits of the tRNA-splicing endonuclease complex, a complex responsible for identification and cleavage of the splicing sites in pre-tRNA [52]. Furthermore, comparing the proteins modulated in the two hyperaccumulators with another work on *A. thaliana* treated with CdS QDs, but where the proteins were separated by 2D-liquid chromatography (LC) with ProteomeLab™ PF2D (Beckman) [53], we found other commonly modulated proteins. We found 2 proteins in common with *A. halleri*: Glutathione S-transferase U24 (Gstu24) and Putative defensin-like protein 121(Lcr55). In common with *N. caerulescens*, we found 3 proteins: Arogenate dehydratase/prephenate dehydratase 6, chloroplastic (Adt6), calcium-binding protein CML31, (Cmlr31), and E3 ubiquitin-protein ligase ARI9 (Ari 9).

Comparing three types of plants of the same Brassicaceae family, one nonmetal hyperaccumulator and two hyperaccumulators, with a low throughput but reliable methodology 2D SDS-PAGE, the total number of proteins modulated in *A. thaliana* [20] was larger than in the two hyperaccumulators (98 vs. 43 and 61). Therefore, the regulative response of *A. thaliana* was stronger than in the two hyperaccumulators in response to Cd or Zn. For example, the pathways, mitochondrial electron transport and ATP synthesis, hormone metabolism, abiotic and biotic stress, and redox ascorbate and glutathione biosynthesis, were almost absent in the two hyperaccumulators. This may be related to the fact that the two hyperaccumulators resist better to exposure as reported [7] and do not need an extensive remodeling of gene expression; they are substantially “tolerant” to the metals in nano or ionic form.

## 5. Conclusions

In conclusion, gene regulation at the level of proteins produced in both hyperaccumulators was differentially affected by exposure to ionic or nano forms of Cd or Zn. Additionally, in the hyperaccumulator plants, metal uptake, transport, and accumulation of metals inside plant are energy-demanding processes, and here there was a confirmation of this in the general modulation of photosynthesis-related proteins observed also in Farinati et al., (2009) [22]. In fact, several proteins regulated by exposure belong to *ATP-dependent activity* in GO analysis. Another pathway and group of proteins, regulated in the two hyperaccumulators, are those related to ROS production, with stress response and with chaperoning of damaged proteins. The final conclusion concerns the difference between ionic and nano-metals, both Cd and Zn. This was already shown at a phenotypic [7] and transcription level [38], but it is confirmed here at a protein level. The mechanism of hyperaccumulation certainly extends from ionic to nano-metals, as well the mechanism of resistance [7]. However, the mechanism of tolerance rests on different proteins and functions, not excluding those that lead to a biotransformation of the nano-form once inside the plant [39].

## Figures and Tables

**Figure 1 nanomaterials-12-04236-f001:**
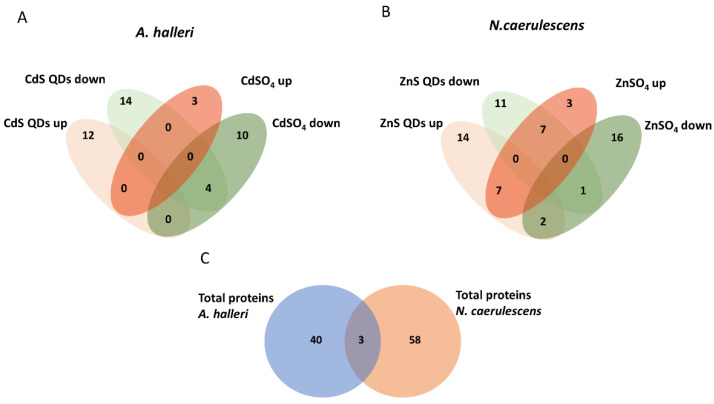
Representation of the effect of exposure to Cd (CdS QDs or CdSO_4_) and Zn (ZnS QDs or ZnSO_4_) on proteome. The Venn Diagrams stand for up and down-regulated proteins treated with Cd in *A. halleri*: (**A**) and treated with Zn in *N. caerulescens* (**B**). In (**C**), proteins in common between two hyperaccumulators are numerated.

**Figure 2 nanomaterials-12-04236-f002:**
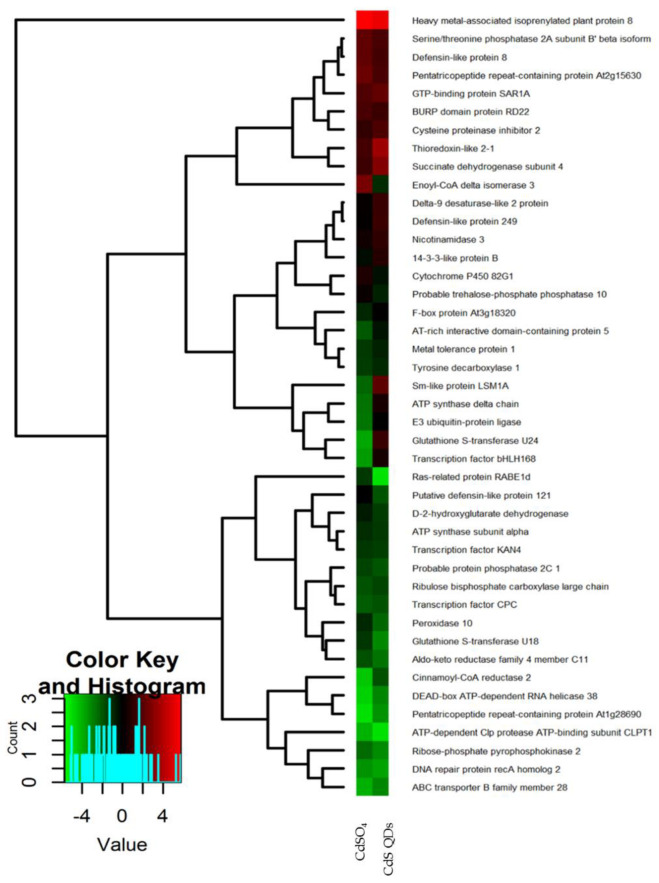
Heat map of proteins of *A. halleri* identified after separation by 2D SDS-PAGE and obtained through the “R” software. The columns represent two different treatments, while the rows represent the identified proteins. The shade of red indicates high abundance of proteins and shade of green indicates low abundance of proteins in the treated samples as compared to the control, which therefore is not shown here. In particular, the first column represents the abundance of proteins in the treated samples with CdSO_4_ and the second column represents the abundance of proteins in the treated samples with CdS QDs. The heat map was assembled by cluster analysis therefore similar levels of expression in treatments as compared to the control are positioned at short distance, while different levels of expression in treatments as compared to the control are positioned at long distance. The more abundant proteins in treated samples compared to the control are positioned at the top, the less abundant proteins are positioned at the bottom. Proteins that have an opposite behavior in the treated samples and in the control are positioned in the center of the figure.

**Figure 3 nanomaterials-12-04236-f003:**
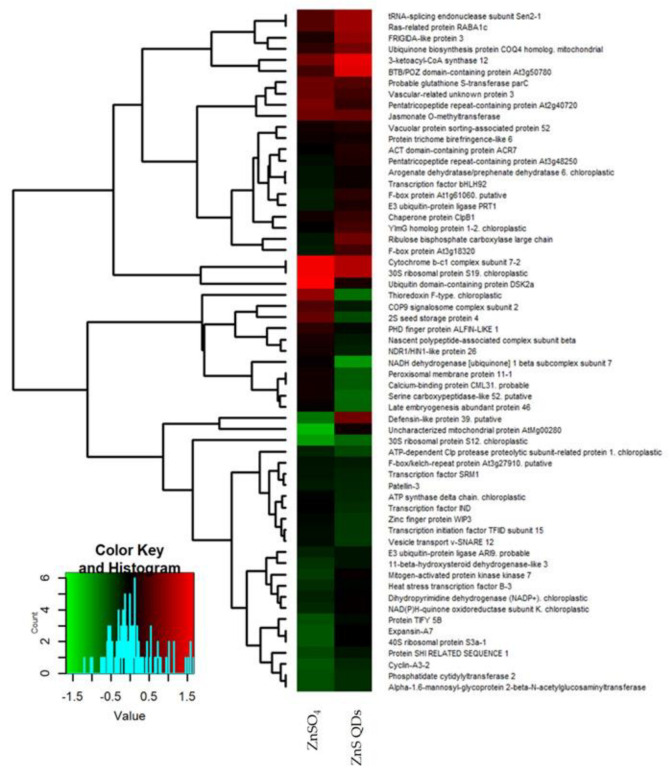
Heat map of proteins of *N. caerulescens* identified after separation by 2D SDS-PAGE and obtained through the “R” software. The columns represent two different treatments, while the rows represent the identified proteins. The shade of red indicates high abundance of proteins and the shade of green indicates low abundance of proteins in the treated samples as compared to the control which therefore is not shown here. In particular, the first column represents the abundance of proteins in the treated samples with ZnSO_4_ and the second column represents the abundance of proteins in the treated samples with ZnS QDs. The heat map was assembled by cluster analysis therefore similar levels of expression in treatments as compared to the control are positioned at short distance, while different levels of expression in treatments as compared to the control are positioned at long distance. The more abundant proteins in treated samples compared to the control are positioned at the top, the less abundant proteins are positioned at the bottom. Proteins that have an opposite behavior in the treated samples and in the control are positioned in the center of the figure.

**Figure 4 nanomaterials-12-04236-f004:**
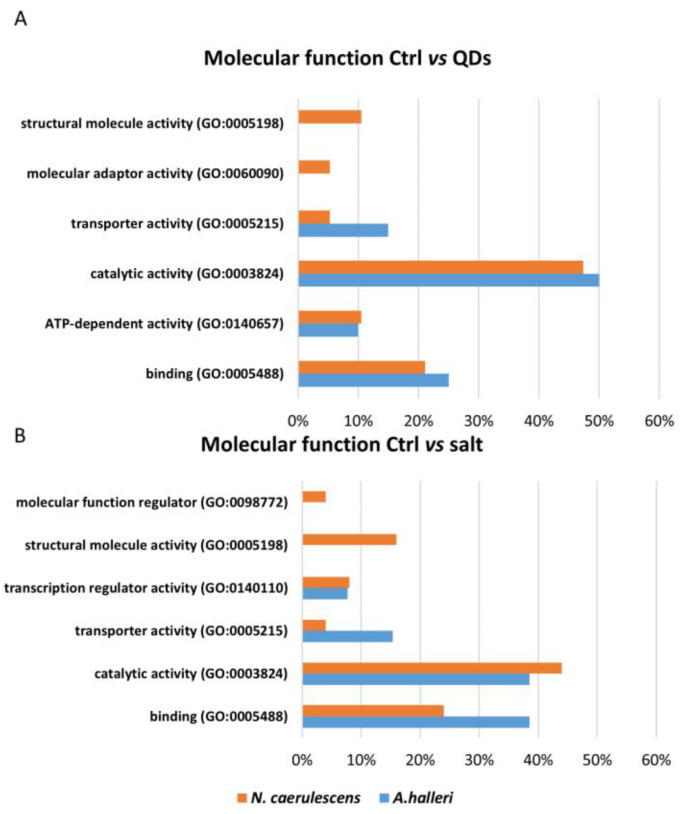
The GO analysis of molecular function. The genes list inserted are genes that code for proteins identified using comparative pairwise proteomic analysis between control, treated with metal QDs and metal salt. Blue bars represent the percentage of the proteins found in *A. halleri*, while orange bars represent the percentage of the proteins found in *N. caerulescens*.

**Figure 5 nanomaterials-12-04236-f005:**
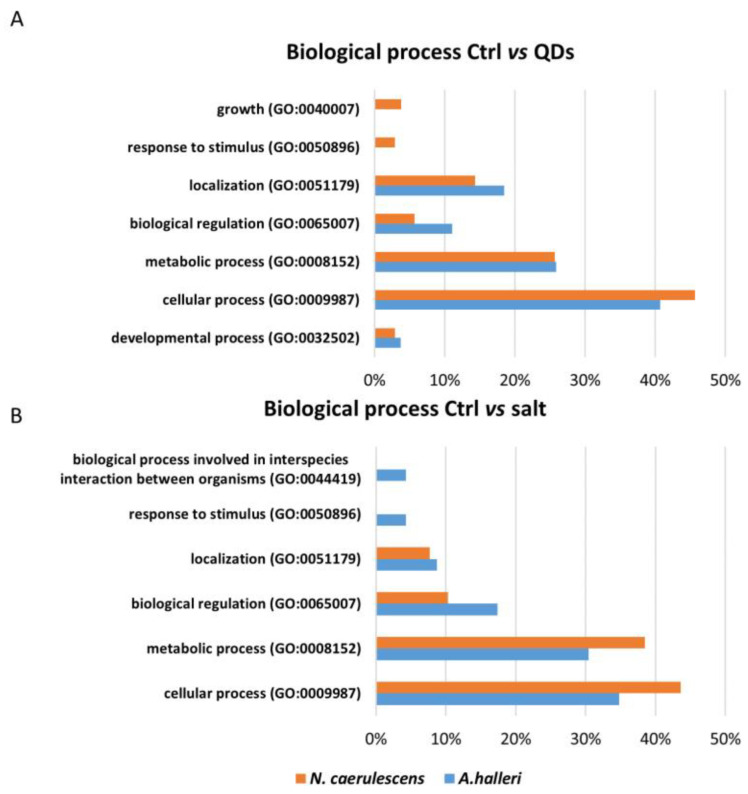
The GO analysis of biological process. The genes list inserted are genes that code for proteins identified using comparative pairwise proteomic analysis between control, treated with metal QDs and metal salt. Blue bars represent the percentage of the proteins found in *A. halleri*, while orange bars represent the percentage of the proteins found in *N. caerulescens*.

**Figure 6 nanomaterials-12-04236-f006:**
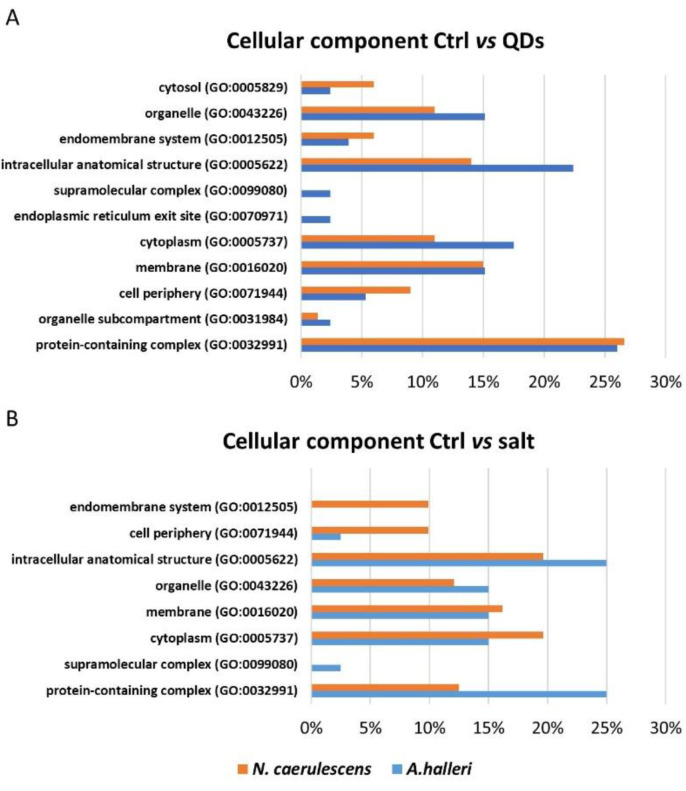
The GO analysis of cellular components. The genes list inserted are genes that code for proteins identified using comparative pairwise proteomic analysis between control, treated with metal QDs and metal salt. Blue bars represent the percentage of the proteins found in *A. halleri*, while orange bars represent the percentage of the proteins found in *N. caerulescens*.

**Figure 7 nanomaterials-12-04236-f007:**
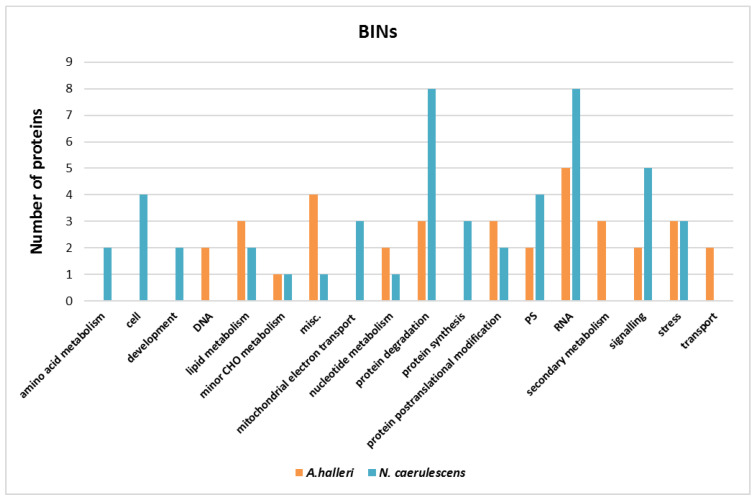
Responsive proteins according to MapMan ontology. The distribution of responsive proteins in *A. halleri* (in blue) and of *N. caerulescens* (in orange) according to MapMan ontology classification (BINs). On the x axis there are the different MapMan bin types, and on the y axis there are the number of proteins for each bin. The data reported are for the nano and ion forms of both metals pooled together for the two plant species.

## Data Availability

Not applicable.

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
