# Peer review of "Protein Analysis of A. halleri and N. caerulescens Hyperaccumulators When Exposed to Nano and Ionic Forms of Cd and Zn"

_nanomaterials, 2022, doi:10.3390/nano12234236_

Round 1
Reviewer 1 Report
Please, see the document attached.

Author Response
Answers to Reviewer 1
Reviewer 1
COMMENTS TO Protein analysis of A. halleri and N. caerulescens 2 hyperaccumulators when
exposed to nano and ionic forms of 3 Cd and Zn.
The article describes the different protein expression levels quantified by 2D-SDS-PAGE in 2
Brasicacee species in 2 metals (Zn and Cd) with various concentrations and forms (nano and
ionic) compared to a "non-stress" situation. The study is quite well designed and the
comparison are adequate, although taking into account the low number of proteins
differentially expressed and the low overlap between the comparison, maybe other analysis
could be also performed.
The comment is quite relevant. There are two different items that were considered. First, the low number of proteins differentially expressed. It was clearly stated along the manuscript (lines 361-3677) that the method of protein separation 2D- PAGE was not a high throughput proteomic method like it may had been iTRAQ as we have used in another paper (Gallo et al., 2020). In that paper we have compared the two methods (2D-PAGE and iTRAQ), and differences in the output were paramount.
Moreover, the staining of the proteins on the 2D-gel was performed with a like Blue Coomassie staining, slightly improved, but with a sensitivity range within nanograms (20-50 ng) of proteins. This was both good and bad. Only proteins whose relative abundance was above the detection limit of the staining were detectable and passed on to the MS identification. This gave the certainly that we were following major changes consequent to a phenomenon which possibly also requests these major changes. Minor changes were not detectable with this method as they could have been with iTRAQ (again in Gallo et al., 2020). Silver-staining is much more sensitive (detection limit ~2 ng) than Coomassie and detects more proteins, but this could had made proteins isolation for MS more troublesome because silver stains use either glutaraldehyde or formaldehyde as the enhancer. These reagents can cause chemical crosslinking of the proteins in the gel matrix, limiting compatibility with destaining and elution methods for analysis by mass spectrometry (MS). Therefore, optimization of sensitivity vs. protein recoverability is critical when employing silver staining as part of an MS workflow (Dyballa and Metzger 2009).
Second, the relatively low number of overlapping in protein up or down regulated between A. halleri and N. caerulescens. We performed the separation and identification several times, but we come always to the same results. We do not exclude again that using iTRAQ in both cases, we could have more similarities, at low, very low, concentration range. But this is not what we wanted to know. We were searching for those major changes which can make sense to a molecular data which is consequent to large phenotypic difference between two plants which at the end converge to a similar evolutionary acquisition: metal tolerance and metal hyperaccumulation. So, we were quite happy that the differences were many and the similarities only few. Considering also what reported in another paper a physiological and morphological differences between A. halleri and N. caerulescens (Imperiale et al., 2022) the low number of overlapping in protein regulation between the two, makes even sense. The same low number of overlapping proteins between the two proteomes is evident from the papers of Visioli et al.; (2012) and Farinati et al., (2009).
The discussion could be improved, as they are quite descriptive of
the results, but it does not go deeper in the main differences.
We thank the reviewer for these important observations. We have modified the discussion trying to go more deeper into these parts as suggested by the reviewer. We have focalized the discussion on the main differences and pointed out more clearly the meaning of these differences and the impact that they have on the physiology of the hyperaccumulation and metal hyper-tolerance. Continuous references were reported to the differences between the nano and the ionic forms of the same metal and the repercussions that these have on the proteome of the two plant species A. helleri and N. caerulescens.
Lines 373-392: “Considering the type of staining used, the concentrations of proteins estimated for the spots selected ere in the order of the ng. In this paper we were searching for those major changes which can make sense to the phenotypic differences between the two species of plants which at the end and converge to a similar evolutionary acquisition: metal tolerance and metal hyperaccumulation. The spots shared between treatments in the pairwise comparisons were displayed in a Venn Diagram as in the figure 1. The spots relevant from a biological point of view were then identified by MS and assigned to a specific molecular function, metabolic pathway, and cellular component. Considering the different origins of the plant species, from a laboratory seed collection for A. halleri, and from a natural site for N. caerulescens, and the different treatments administered, Cd and Zn, respectively, it was not surprising that there was a low overlapping in the number of regulated proteins found. Besides, this result was already evident from the two papers [22,24].
The two different pH ranges used for the first-dimension gel in the two hyperaccumulators respected the separation patterns that we observed at all pH ranges tested. The range between the two were 5-8 for A. halleri and 3-10 for N. caerulescens. After a fine-tuning of the protocol, the same pH range was used for A. halleri as for A. thaliana (pH 5-8) [20], as it respected the same protein profile. On the other hand, N. caerulescens seemed to have a more extensive protein profile towards the acidic ph part, for this reason a more extensive ph range was used (pH 3-10).”
Lines 405-408: “The downregulation of GSTFs by metals, was observed also in another study, and requires further investigation because GSTs are usually induced under metal stress [30]. Similar evidence and a similar conclusion are reported in Farinati et al., (2009) [22].
Lines 413-426: “It is of particular importance to have found proteins related to the trichome because in literature there are evidence that Cd in A. halleri is mainly stored in the leaf trichomes [33]. Moreover, Zhao et al., (2006) [34] found that A. halleri hyperaccumulate Cd and that the root-to-shoot translocation of Cd was in a certain way inhibited by Zn, hinting that Cd enters A. halleri cells partly through the Zn transport pathway. Schvartzman, et al., (2018), propose that transcription factors, as HMA4 was selected during evolution of A. halleri and contributes to metal tolerance by enabling metal storage in shoot tissues. In addition, there are evidence that Metal Tolerance Protein 1 (Mtp1) is probably responsible for Zn storage in shoot vacuoles, attributing Zn hyper tolerance, but can also transport cadmium with a low efficiency [35]. The protein Mtp1, in this proteomic analysis, was upregulated in two treatments of A. helleri, therefore, it seems important that we found this protein as proteins modulated in A. halleri by the treatments, a transcription factors and a metal-chaperones that may serve for the translocation and the storage of the metals in the above parts of the plants. “
Lines 437-439: It is evident that the regulation imposed by ZnS QDs and CdS QDs required more ATP, possibly because it is more complex [37, 38].
Lines 480-482:” Part of these proteins were already described in Visioli et al., (2012), though in a different context, because there were used different extraction method and the study was on metal ions only [24].”
Lines 496-499: “Also, in the previous study [24] there was a report of several proteins involved in energy metabolism (both in chloroplast and mitochondrion) that were in-creased. Therefore, they can compensate for a reduction in ATP synthase.”
Lines 529-546: “Also, the presence of Ribulose bisphosphate carboxylase large chain (Rubisco subunit) corresponded to a higher rate of photorespiration. In fact, the photorespiratory pathway represents a way to recover C atoms ‘‘wasted’’ by this side activity of Rubisco. The cycle requires the coordinated activity of chloroplasts, peroxisomes, and mitochondria. In N. caerulescens there are three proteins that are part of the mitochondrial electron transport and ATP synthesis pathway: Ubiquinone biosynthesis protein COQ4 homolog (COQ4), and Cytochrome b-c1 complex subunit 7-2 (QCR7-2) and NADH dehydrogenase [ubiquinone] 1 beta subcomplex subunit 7 (At2g02050). The first two proteins were up regulated after both treatments, while At2g02050 was down regulated in the treatment with ZnS QDs and up regulated in the treatment with ZnSO4 (Figure 3). Conversely in A. halleri no proteins belonging to this pathway showed a particular regulation after treatments. These data, together with other present in literature, suggest that the hyperaccumulation of heavy metals in shoots of hyperaccumulator plants is energetically extremely costly, so the plants can afford this cost by reinforcing the photosynthetic mechanism modulating proteins involved in photosynthesis and those components that act as scavengers of the unavoidable by-products that occur with this extensive energy handling [22, 24]. This phenomenon was also explained in terms of “trade-off” between mass growth and hyper-tolerance in Maestri et al.; (2010) [52].
Suggestions to the article:
A short description of the plant phenotype could be included (It seems, by the images
in the supplemental material, that the treated plants are bigger).
We thank the reviewer for this important observation. As reported in the papers (Imperiale et al., 2022; Visioli et al., 2012 and Visioli et al., 2010) different types of hyperaccumulators as A halleri and N. caerulescens are not only metal tolerant, but also, they need metal to grow and be healthy. Therefore, it was not surprising that after the treatments the plants were bigger and better than when they were untreated, or in comparison to the controls. The description of the plant phenotype is in the captions to Figure S1 and S2 of the Supplementary Materials. We also have inserted a short paragraph in the text (3.1) to describe them for the reviewer.
Lines 172-179: “In a previous paper, it was shown that samples of Noccea caerulescens from Monte Prinzera varied according to the subsites and may give plants with different phenotypes according to the subsite [24]. The plant we used where taken from the same subsite, number 3, as were those in the paper from Imperiale et al., (2022) [7], where a thorough description is provided. These plants, as those of Arabidopsis halleri, not only grow well in the presence of Cd and Ni at high concentrations, that may affect negatively other plants non hyperaccumulators, but also, they seem to need high concentrations of metals to grow more vigorously, as shown in Figure S1 and S2.”
As mentioned, the genomes of A. halleri and N. caerulescens are 88.5% and 94%
similar to those of A. thaliana. Maybe the difference with A. halleri is quite high to
assume that this is not going to affect the results. It is also known the similarity at
proteomic level? An explanation of this could be added, due to the classification of the
proteins was done using MapMAn software base on A. thaliana. I guess there are not
specific databases for each specie, aren’t they?
We thank the reviewer for the comments. In fact, the difference in the genomes of the two hyperaccumulators is significant and this can certainly weight on the differences we have found in protein regulation. Whereas the proteome of A. halleri was quite studied not much evidence have been reported for N. caerulescens. The classification of the proteins based on A. thaliana proteome is indicative, as it is close in similar cases such as those reported in [24].
MapMan is a software tool for gene functional annotations in plant sciences. It was developed to facilitate improvement, consolidation, and visualization of gene annotations across several plant species. In particular, GoMapMan is based on the MapMan ontology, organized in the form of a hierarchical tree of biological concepts, which describe gene functions. Currently, genes of the model species Arabidopsis thaliana and only three crop species (Solanum tuberosum, Solanum lycopersicum and Oryza sativa) are included (Ramsak et al., 2013).
Therefore, the use of MapMan allowed a better description of the differences among species and treatments, leading us to final interpretations in 3.4 Pathway analysis and in 4.3 Pathway analysis
Why is using only the aerial part of the plant? Why is not also collected also the roots
(when is supposed the most amount of the metal should be accumulated, and the
changes in the proteome could be higher)? Or do the same study by using the roots,
and expanded the comparison between aerial part of the plant and the roots?
We thank the reviewer for these important observations. The use of the aerial part of the plant for the proteomic analysis was consequent to the discovery that in hyperaccumulators and the largest part of the metal accumulation occurs by definition in the aerial parts (Baker and Whiting 2002. In Search of the Holy Grail: A Further Step in Understanding Metal Hyperaccumulation? New Phytologyst https://www.jstor.org/stable/pdf/1513875.pdf?refreqid=excelsior%3A96b8e1ddb62cbcb2a56d1b3139eff709&ab_segments=&origin=&acceptTC=1; and in Van der ent 2013, https://www.frontiersin.org/articles/10.3389/fpls.2015.00554/full).
Particularly in A. halleri and N. caerulescens the metal/nano metal accumulate into the aerial part at the largest concentrations (Imperiale et al., 2022). The aerial parts are therefore the tissues more exposed to the metal stress and in which the hyperaccumulation in certain tissues had to deal with the necessity of resisting the inhibitory action of the same metals in ionic or nanoforms. Paradigmatic at this propose has been the discovery that “biotransformation” event might be involved in this phenomenon (Marmiroli et al., 2022). Previous proteomic studies on A. halleri (Farinati et al., 2009) and on N. caerulescens (Visioli et al., 2012) also used only shoots for their protein extractions.
Comparisons were made between each treatment and the control and the results of
both are aggregated to compare between them, but the overlapping is quite low. It is
possible to compare the same metal different treatments between them? This could
show if the form (nano or ionic) of metal treatment affects the plants directly.
The comparison in Figures 2 and 3 were indeed between treatments (ionic and nano) within the same plant species. Figure 1 summarize all the treatments within the same species but also between the two species and the number of changes were there reported in Figure 1C. It is evident, both from Figure 1, 2 and 3, that in both species’ nano and ionic form of the same metal (Cd or Zn) affect the plants directly but in different way.
Which is the meaning of the opposite behavior of some proteins in the same metal
treatment between forms (as ATP synthase delta chain in A. halleri)?
Metals in their ionic or nano form determine different responses in the plant. This was shown at morphological/ physiological level (Imperiale et al., 2022), at transcriptomic level (Marmiroli et al., 2014) and at proteomic level (Marmiroli et al., 2015 and Gallo et al., 2021). Difference like the opposite behavior of ATP synthase delta chain in A. halleri falls within this vision. A recent study of A. thaliana treated with different nanomaterials has shown that an epigenetic response resulting in the differential replication/ repair of segments of the mitochondrial and chloroplastic genome were involved (Pagano et al., 2022). The purpose seems to be the regulation of mitochondrial/chloroplastic protein assembly including therefore the ATP synthetase. Curiosity is that the epigenetic regulation was different when triggered by nano or ionic forms. Comments on this point were reported in lines 432-439:” It is evident that the treatment with CdS QDs determines the modulation, up or down, of proteins involved in oxidative phosphorylation as the ATP-dependent activity class listed in Figure 4A. In fact, Gallo et al., (2020) [37] demonstrated that CdS QDs treatments caused a reduction in oxidative phosphorylation and in ATP production in yeast. Moreover, some proteins, involved in the energy production, ATP synthase subunit alpha (Atp1) and ATP-dependent Clp protease ATP-binding (Clpt1), were down regulated (Figure 2). It is evident that the regulation imposed by ZnS and CdS QDs required more ATP, possibly because it is more complex [37, 38]”
And in lines 482-498: “Concerning GO analysis, in the molecular function bar chart, the ATP-dependent activity class is shown in the comparison Ctrl vs QDs (Figure 4A). Two proteins involved in energy production, ATP synthase delta chain chloroplastic (Atpd) and ATP-dependent Clp protease proteolytic subunit-related protein 1 chloroplastic (Clpr1), were down regulated (Figure 3) as reported for A. halleri. These two proteins were localized together in the chloroplast, and previous studies in A. thaliana treated with QDs highlighted the role of chloroplast as a potential target of the ENMs [20, 44]. Moreover, it is reported that mutation in a gene encoding for a protein of chloroplast function led to resistance to QDs [40]. Clpr1was required for chloroplast development and differentiation [45]. Atpd is an ATP synthase which produces ATP from ADP in the presence of a proton or sodium gradient and is essential for photosynthesis, by facilitating electron transport in both photosystems I and II [46]. Wang et al., (2016) showed that CuO NPs impedes electron transport between the two photosystems which can cause an increase of ROS and generation of oxidative stress, damaging biological molecules and disrupting cellular metabolism [47]. Also, in the previous study [24] there was a report of several proteins involved in energy metabolism (both in chloroplast and mitochondrion) that were increased. Therefore, they can compensate for a reduction in ATP synthase.”
In my opinion the comparison with article [20] in point 4.4 is not quite adequate, not
the experimental conditions neither the type of plant is the same. In [20] all plants
were collected for protein extraction including roots, that could be the part of the
plant that most accumulate the metals.
Comparison with Gallo et al., (2021) was made only for the protein related with A. thaliana w.t., not for the tolerant mutants. The condition of the treatment with CdS QDs were quite similar to those reported in the manuscript for A. halleri and N. caerulescens. It is correct to note that in that case both root and leaves were collected for protein analysis. The fact that in A. thaliana the level of translocation from roots to shoots is significantly lower than the root to shoot translocation in A. halleri and N. caerulescens (Imperiale et al., 2022), support the observation of the reviewer. However, though the proteome in this case were a mix between those of roots and shoots, nevertheless the comparison can have a sense. In addition, as stated above in answer number 3, for the hyperaccumulators A. halleri and N. cerulescens the roots were not the tissue mostly stressed by the metals’ treatment because both forms of metals, nano and ionic, were translocated to the aerial parts of the plants efficiently (Imperiale et al., 2022). To the same conclusion arrived also Farinati et al. 2009 and Visioli et al 2012.
Suggestions to the text:
Lines 58-62, could be rewrite to be more understandable?
The text has been modified: Noccea caerulescens (former Thlaspi caerulescens) and Arabidopsis helleri have been model plants in metal hyperaccumulation studies [7, 11], largerly because of their genetic analogy with the model species Arabidopsis thaliana (L.) Heynh and because their genomes have 94% and 88.5% correspondence in coding regions respectively [12].
Line 119, pH range for A. halleri. Although in line 380 there is a kind of justification,
could be more explained as it seems too much restrictive in comparison to the pH
range (3-10) of N. caerulescens.
It is evident from the literature (Farinati et al, 2009) that other authors utilized a 3-10 pH range also for A. helleri found that most of the proteins fell in the pH range 5-8. Therefore, we added in the text:
Lines 395-399: “The pH range between the two were 5-8 for A. halleri and 3-10 for N. caerulescens. After a fine-tuning of the protocol the same pH range was used for A. halleri as for A. thaliana (pH 5-8) (Gallo et al 2021) as it respected the same protein profile. On the other hand N.caerulescens seemed to have a more extensive protein profile towards the acidic pH part (24), for this reason a more extensive pH range was used (pH 3-10).”
Line 159. On extra space.
We eliminated the extra space
Line 283. It is written in figure 3, 4 and 5, but I think it should be 4, 5 and 6.
We changed the numbers of the Figures in the text in “Figure 4,5,6”
Lines 341-349. The text does not match exactly with the legend in figure 3, so it not
easy to follow it.
We recast the lines (now 338-345): “In A. halleri, the set of reprogrammed proteins was associated with the following major MapMan bins: RNA regulation of transcription, secondary metabolism, miscellaneous (misc.), protein post-transcriptional modification, protein degradation, and lipid metabolism (Figure 7). In N. caerulescens, the set of reprogrammed proteins was associated with protein degradation, RNA regulation of transcription, photosynthesis and photorespiration (PS), signaling, and cell (Figure 7). This last pathway, together with amino acid metabolism, and development, are exclusive of N. caerulescens, while DNA synthesis, secondary metabolism and transport are exclusive of A. halleri (Figure 7).”
Something that takes my attention is that there is not an increased expression of
metallothionein proteins (https://www.uniprot.org/uniprotkb/P43392/entry), that are small proteins (around 6.5KDa, but it depends a lot between species) cys-enriched (around 30%) that
is suggested to be Zn and Cd accumulators... maybe the low MW of these proteins difficult the
detection on the 2D SDS-PAGE.
Conventional 2D- PAGE is restricted to the detection of denatured proteins in the size range of 10~200 kDa (Saraswathy et al., 2011. Concepts and Techniques in Genomics and Proteomics, Book). Small proteins are not easy to detect, and they were not detected neither by Farinati et al. 2009, but by Visioli et al 2012 using HPLC-MS method.
The conclusion is ok regarding the results presented.
Why is so different the expression between the ionic/nanoforms as the metal quantity is the same? The authors could hypothesize a little bit on this, maybe the bioavailability of the metal within the forms???
We have discussed this point in several instances in the text when comparing proteins differences between species and treatments. We also reported several references in which more data on the difference between nano and ionic form of a metal were reported at all levels, phenotypic (Imperiale et al. 2022), transcriptomics (Pagano et al., 2018, https://www.sciencedirect.com/ science/article/pii/ S2468584418300783 ; Marmiroli et al 2021, https://pubs.acs.org/doi/full/10.1021/acs.est.1c01123 ), micro-RNA regulation (Pagano et al 2021). Hypothesis and mechanisms were formulated in a recent review by our group (Pagano et al 2022 in press). However, there is not a clear consensus on this difference. Certainly, the fact that the nanoparticles are more energetic from a structural point of view than the salt, plays a major role in the difference.
- Imperiale, D.; Lencioni, G.; Marmiroli, M.; Zappettini, A.; White, J.C.; Marmiroli, N. Interaction of hyperaccumulating plants with Zn and Cd nanoparticles. Science of the Total Environment 2022, 817, 152741
- van der Ent, A.; Baker, A.J.M.; Reeves, R.D.; Pollard, A.J.; Schat, H. Hyperaccumulators of Metal and Metalloid Trace Elements: Facts and Fiction. Plant Soil 2013, 362, 319–334.
- Imperiale, D.; Lencioni, G.; Marmiroli, M.; Paesano, L.; Zappettini, A.; White, J.C.; Marmiroli, N. Data on the Interaction of Hyperaccumulating Plants with Nanoscale Metals Zn and Cd. Data Br. 2022, 42, 108171
- Farinati, S.; Dal Corso, G.; Bona, E.; Corbella, M.; Lampis, S; Cecconi, D.; Polati, R.; Berta, G.; Vallini,G.; Furini, S.;Proteomic analysis of Arabidopsis halleri shoots in response to the heavy metals cadmium and zinc and rhizosphere microorganisms. Proteomics 2009, 9, 4837–4850.
- Visioli, G.; Vincenzi, S.; Marmiroli, M.; Marmiroli, N. Correlation between phenotype and proteome in the Ni hyperaccumulator Noccaea caerulescens caerulescens. Enviromental and Experimental Botany 2012, 77, 156-164.
- Kupper, H.; Kochian L.V.; Transcriptional regulation of metal transport genes and mineral nutrition during acclimatization to cadmium and zinc in the Cd/Zn hyperaccumulator, Thlaspi caerulescens(Ganges population). New Phytologist, 2010, 185, 114-129.
- Zhao, F. J.; Jiang, R. F.; Dunham, S. J.: McGrath, S.P. Cadmium uptake, translocation and tolerance in the hyperaccumulator Arabidopsis halleri. New Phytologist, 2006, 172, 646-654.
- Schvartzman, M.S.; Corso, M.; Fataftah, N.; Scheepers, M.; et al., Adaptation to high zinc depends on distinct mechanisms in metallicolous populations of Arabidopsis halleri. New Phytologist, 2018, 218, 269–282
- Gallo, V.; Srivastava, V.; Bulone, V.; Zappettini, A.; Villani, M.; Marmiroli, N.; Marmiroli, M. Proteomic analysis identifies markers of exposure to Cadmium Sulphide Quantum Dots (CdS QDs). Nanomaterials 2020, 10, 1214.
- Pagano, L.; Marmiroli, M.; Villani, M.; Magnani, J.; Rossi, R.; Zappettini, A.; White, J.C.; Marmiroli, N. Engineered Nanomaterial Exposure Affects Organelle Genetic Material Replication in Arabidopsis Thaliana. ACS Nano 2022, 16, 2249–2260.
- Marmiroli, M.; Pagano, L.; Rossi, R.; Torre-roche, R.D. La; Lepore, G.O.; Ruotolo, R.; Gariani, G.; Bonanni, V.; Pollastri, S.; Puri, A.; et al. Copper Oxide Nanomaterial Fate in Plant Tissue: Nanoscale Impacts on Reproductive Tissues. Sci. Technol. 2021, 10769–10783.
- Maestri, E.; Marmiroli, M; Visioli, G.; Marmiroli, N. Metal tolerance and hyperaccumulation: Costs and trade-offs between traits and environment. Enviromental and Experimental Botany 2010, 68, 1-13.
- Saraswathy et al., 2011. Concepts and Techniques in Genomics and Proteomics, Book).
- Pagano et al., 2018, https://www.sciencedirect.com/ science/article/pii/ S2468584418300783

Reviewer 2 Report
- The data is presented in a clear and organized way, with some grammar and spelling details to improve. The results are technically sound.
- Check spelling throughout the main document, SI, and figure labels; the use of upper, lower case, and subscripts.
- If you have the elemental analysis, would it be possible to include where Cd or Zn are going in the plants from the salts or the QDs?
- Mats and methods – Do you have CdS dry diameter and ZnS density?
- Parts of the discussion seem to be presented as results and need to explain or interpret rather than describe the finding. For example lines 385-399, what is the meaning? If Cd is interfering the detox process and more so the nano-form, what is the plant doing to Cd instead? Can you show or speculate if the plant is sending the metal from the salts or from the QDs to different locations?
- Lines 521-531 – Good discussion
Author Response
Answers to Reviewer 2
Reviewer 2
Comments and Suggestions for Authors
Question 1
- The data is presented in a clear and organized way, with some grammar and spelling details to improve. The results are technically sound.
Answer 1
We thank the Reviewer for the comment. We went through the manuscript and corrected the spelling and grammar mistakes.
Question 2
- Check spelling throughout the main document, SI, and figure labels; the use of upper, lower case, and subscripts.
Answer 2
We thank the reviewer for the observation. We did as we were instructed and corrected the mistakes as far as we could see them.
Question 3
- If you have the elemental analysis, would it be possible to include where Cd or Zn are going in the plants from the salts or the QDs?
Answer 3
We thank the reviewer for this question. In another paper that our group published on the same plants we measured the concentrations of Cd and Zn in all the parts of the plants, and we found out that most of the metals were translocated to the above ground plant parts. This paper (Imperiale, D.; Lencioni, G.; Marmiroli, M.; Zappettini, A.; White, J.C.; Marmiroli, N. Interaction of hyperaccumulating plants with Zn and Cd nanoparticles. Science of the Total Environment 2022, 817, 152741) states that the concentrations in the plant parts of Noccea caerulescens under higher treatment of Zn Quantum dots (30 mg ZnS QDS) had in the roots 413.7mg/kg dw of Zn, translocating to the shoots 2218.5 mg/kg dw of Zn. When treated with the Zn salt (30 mg ZnSO4) they had 851.4 mg/Kg dw and 2610.5 mg/kg dw of Zn in the roots and shoots respectively. When the Zn concentration in the treatment as QDs was lower (10 mg ZnS QDS) the concentration of Zn in the root was mor or less the same 492.7 mg/Kg dw of Zn while in the aerial parts was even higher 2397.5 mg/Kgdw Zn. Also, for the lower treatment of Zn salt (10 mg ZnSO4) the concentration in roots was almost the same: 848.9 mg/Kg Zn and the shoots again was even higher: 3244.7 mg/kg Zn.
Arabidopsis halleri under the higher Cd quantum dots treatment (30 mg CdS QDS) had in the roots 63.8 mg/kg of Cd, while in the aerial parts, under the same treatment showed 75,9 mg/kg dw of Cd. Under the treatment with Cd salt (30 mg CdSO4) the plant had in the roots 65.3 mg/kg dw of Cd and in the aerial parts 134.2 mg /kg dw of Cd. Considering the lower treatment of CdS QDs (10 mg CdS QDS) the concentration of Cd in the roots and in the aerial parts was also lower: 24.8 and 48.6 mg/Kg Cd respectively. Under the lower Cd salt (10 mg CdSO4) the Cd concentrations both in the roots and in the aerial parts were lower: 13.6 mg/Kg and 71.0 mg/Kg of Cd respectively.
Overall, we can conclude that the Zn translocation from roots to shoots in N. cearulescens does depend little on the treatment type and concentration. In A. halleri the root to shoot translocation was lower than in N. caerulescens, possibly also because the higher toxicity of Cd in comparison to Zn, but the difference between roots and shoots was significant (about two).
Since we have already published the concentrations data, we did not want to repeat them here in this manuscript, but just quoted the relevant literature for the case in point.
Question 4
- Mats and methods – Do you have CdS dry diameter and ZnS density?
Answer 4
Also, here we can quote the relevant literature where the quantum dots were fully characterized which is another of our works on quantum dots and nanomaterials: Pagano et al. ACS Nano 2022, 16, 2, 2249–2260. https://pubs.acs.org/doi/full/10.1021/acsnano.1c08367
In that paper there are the diameters of the two QDs which are both below 5nm, and the percentage of Cd in CdS QDs which is 78% Cd and the percentage of Zn in ZnS which is 63.2% Zn. In the same paper there is a full characterization of both types of Quantum dots and their HR TEM images.
Question 5
- Parts of the discussion seem to be presented as results and need to explain or interpret rather than describe the finding. For example, lines 385-399, what is the meaning? If Cd is interfering the detox process and more so the nano-form, what is the plant doing to Cd instead? Can you show or speculate if the plant is sending the metal from the salts or from the QDs to different locations?
Answer 5
In lines 385-399 we report as in the rest of the discussion, the proteins that were influenced by the treatments. It is of particular importance to have found proteins related to the trichome because in literature there are evidence that Cd in A. halleri is mainly stored in the leaf trichomes (Küpper, H., Lombi, E., Zhao, FJ. et al. Cellular compartmentation of cadmium and zinc in relation to other elements in the hyperaccumulator Arabidopsis halleri. Planta 212, 75–84 (2000). https://doi.org/10.1007/s004250000366). Huguet et al (Huguet et al. Environmental and Experimental Botany, 82, 2012, 54-65, https://doi.org/10.1016/j.envexpbot.2012.03.011 ) found that Cd is also stored in the vacuole of the old leaves. This is also consistent with the fact that Cd is extremely toxic. Moreover, Zhao et al (Zhao et al. New Phytologist 2006. https://doi.org/10.1111/j.1469-8137.2006.01867.x) found that A. halleri hyperaccumulate Cd and that the root-to-shoot translocation of Cd was in a certain way inhibited by Zn, hinting that Cd enters A. halleri cells partly through the Zn transport pathway. Schvartzman, et al,2018, (https://nph.onlinelibrary.wiley.com/doi/full/10.1111/nph.14949) propose that the transcription factors, for example such as HMA4 was selected during evolution of A. halleri and contributes to metal tolerance by enabling metal storage in shoot tissues. In addition, they sustain that there is evidence that Metal Tolerance Protein 1 (MTP1) is probably responsible for Zn storage in shoot vacuoles, providing Zn hypertolerance (Schvartzman et al., New Phytologist, 2018, 218: 269–282 doi: 10.1111/nph.14949). Therefore, it seems important that we found as proteins modulated in A. helleri by the treatments, both transcription factors and metallo-chaperones that serves for the translocation and the storage of the metals in the above parts of the plants.The treatment with CdS QDs also can have toxic effects on the plant. A. helleri can also resent the toxicity of CdS QDs, but this was not due to the release of ionic Cd from the QDs (Imperiale et al 2022), but to the typic features of the nanoparticles. The effect of CdS in nano form is exerted by the the uptake that is vesicular rather than with protein transport (ABC transporters). Once inside the plant the QDs interact with the proteins of the cell to form corona proteins (Ruotolo et al 2018, https://pubs.rsc.org/en/content/articlelanding/2018/en/c7en01226h/unauth), which has the role both in intracellular targeting and in toxicity. Inside the cell particular targets of the QDs are the cell organelles (chloroplasts and mitochondria, Pagano et al 2022, https://pubs.acs.org/ doi/full/10.1021/ acsnano. 1c08367). From these interactions derive the more relevant effects like oxidative stress (increase in ROS), inhibition of photosynthesis, activation of stress response. All these changes were reported in terms of specific proteins involved in the Discussion.
The Discussion has been modified and changed in several parts, also to respond to the queries of Reviewer 1, to give more insight in term of hypothesis and explanations. This to answer to the requests to explain and interpret the data. On the other hand, the presentations of some data into the Discussion were made on the ground of the different categories: Molecular Function, Biological Process, and cellular components rather than in terms of raw proteomic differences as described in heat maps and Venn Diagrams. In the section Results on the use of Functional categories was already an attempt to promote or suggest some interpretation. But to comply with the reviewer observations several changes and additions were made to the Discussion
The text has been modified thus:
Lines 402-430: “However, the behavior of glutathione S-transferase (Gstu24) was opposite in the treatment with CdS QDs as compared to treatments with CdSO4, as highlighted in the heat map, whereas the other oxidoreductases were down regulated after the two treatments (Figure 2). The downregulation of GSTFs by metals, was observed also in another study, and requires further investigation because GSTs are usually induced under metal stress [30]. Similar evidence and a similar conclusion are reported in a previous paper [22].
Two transcription factors were down regulated by the treatments: transcription factor CPC (Cpc) and KAN4 (Kan4) (Figure 2). The first is expressed in trichomes and in young developing leaves, it represses trichome development by lateral inhibition [31]; the second regulates carpel integuments formation by modulating the content of flavonols and proanthocyanidin in seeds [32]. It is of particular importance to have found proteins related to the trichome because in literature there are evidence that Cd in A. halleri is mainly stored in the leaf trichomes [33]. Moreover, Zhao et al., (2006) [34] found that A. halleri hyperaccumulate Cd and that the root-to-shoot translocation of Cd was in a certain way inhibited by Zn, hinting that Cd enters A. halleri cells partly through the Zn transport pathway. Schvartzman, et al., (2018), propose that transcription factors, as HMA4 was selected during evolution of A. halleri and contributes to metal tolerance by enabling metal storage in shoot tissues. In addition, there are evidence that Metal Tolerance Protein 1 (Mtp1) is probably responsible for Zn storage in shoot vacuoles, attributing Zn hyper tolerance, but can also transport cadmium with a low efficiency [35]. The protein Mtp1, in this proteomic analysis, was upregulated in two treatments of A. halleri, therefore, it seems important that we found this protein as proteins modulated in A. halleri by the treatments, a transcription factors and a metal-chaperones that may serve for the translocation and the storage of the metals in the above parts of the plants. On the contrary, the gene for heavy metal-associated isoprenylated plant protein 8 (Hipp08), was up regulated in both treatments (Figure 2). Hipp08 is a metallochaperone protein for the safe transport of metallic ions inside the cell [36]. HIPPs may be involved in heavy metal homeostasis and detoxification, particularly concerning Cd tolerance, transcriptional responses to cold and drought, and plant-pathogen interactions [31]”
Question 6
- Lines 521-531 – Good discussion
Answer 6
We thank the reviewer for the positive comment on our work.

Round 2
Reviewer 1 Report
The authors have answered all my questions and have solved all my doubts adequately, no more comments on this.